EDITORIAL

# Bridging Biodiversity and Health: The Global Biodiversity Information Facility's initiative on open data on vectors of human diseases

Paloma Shimabukuro[1,*], Quentin Groom[2], Florence Fouque[3], Lindsay Campbell[4], Theeraphap Chareonviriyaphap[5], Josiane Etang[6], Sylvie Manguin[7], Marianne Sinka[8], Dmitry Schigel[9] and Kate Ingenloff[9]

1 Instituto René Rachou/FIOCRUZ-Minas, Grupo de Estudos em Leishmanioses/Coleção de Flebotomíneos, Avenida Augusto de Lima, 1715, Barro Preto, 30190-009, Belo Horizonte, Minas Gerais, Brazil
2 Meise Botanic Garden, Meise, 1860, Belgium
3 TDR, The Special Programme for Research & Training in Tropical Diseases, World Health Organization, 20 Avenue Appia, 1211 Geneva 27, Switzerland
4 Florida Medical Entomology Laboratory, Department of Entomology and Nematology, IFAS, University of Florida, 200 9th St. SE, Vero Beach, Florida, 32962, USA
5 Kasetsart University, Department of Entomology, Faculty of Agriculture/Director of the Research and Lifelong Learning Center for Urban and Environmental Entomology, 50 Ngam Wong Wan Rd, Lat Yao, Chatuchak, Bangkok, 10900, Thailand
6 Organisation de Coordination pour la lutte contre les Endémies en Afrique centrale (OCEAC) / Faculty of Medicine and Pharmaceutical Sciences (FMPS), University of Douala, Cameroon / Director of Academic Affairs and Cooperation, University of Bertoua, Cameroon
7 Institut de Recherche pour le Développement France-Sud, Uniformed Services University of the Health Sciences, Université Montpellier Faculté des Sciences de Montpellier, Université de Montpellier, 163 rue Auguste Broussonnet, 34090, Montpellier, France
8 University of Oxford, Oxford Long-Term Ecology Laboratory, Department of Plant Sciences, South Parks Road, Oxford, OX1 3RB, UK
9 Global Biodiversity Information Facility (GBIF), Secretariat, Universitetsparken 15, DK-2100, Copenhagen Ø, Denmark

**Submitted:** 20 March 2024

* Corresponding author. E-mail: pshimabukuro@gbif.org

Preprint submitted at https://doi.org/10.5281/zenodo.10843586

Included in the series: *Vectors of human disease* (https://doi.org/10.46471/GIGABYTE_SERIES_0002)

## ABSTRACT

There is an increased awareness of the importance of data publication, data sharing, and open science to support research, monitoring and control of vector-borne disease (VBD). Here we describe the efforts of the Global Biodiversity Information Facility (GBIF) as well as the World Health Special Programme on Research and Training in Diseases of Poverty (TDR) to promote publication of data related to vectors of diseases. In 2020, a GBIF task group of experts was formed to provide advice and support efforts aimed at enhancing the coverage and accessibility of data on vectors of human diseases within GBIF. Various strategies, such as organizing training courses and publishing data papers, were used to increase this content. This editorial introduces the outcome of a second call for data papers partnered by the TDR, GBIF and GigaScience Press in the journal *GigaByte*. Biodiversity and infectious diseases are linked in complex ways. These links can involve changes from the microorganism level to that of the habitat, and there are many ways in which these factors interact to affect human health. One way to tackle disease control and possibly elimination, is to provide stakeholders with access to a wide range of data shared under the FAIR principles, so it is possible to support early detection, analyses and evaluation, and to promote policy improvements and/or development.

**Subjects** Ecology, Biodiversity, Global Health

## INTRODUCTION

Collection, access, and appropriate use of primary biodiversity data are crucial for research on human health and pathogens, as well as interventions to mitigate, control, and eventually eliminate the vectors and the pathogens they transmit. There is an increased interest in primary biodiversity data related to vectors, hosts/reservoirs, and pathogens due to the numerous outbreaks of emerging and-re-emerging zoonotic diseases [1, 2].

Multiple factors, including climate change, transportation, anthropogenic landscape disturbance, land use change with examples such as deforestation and mining encroachment on wild areas are impacting the distribution of vector borne diseases (VBDs) with several emerging and re-emerging pathogens expanding in geographic range [3–5]. When an emergent or re-emergent disease causes an outbreak, access to data becomes critical for a rapid and effective public health response. Through GBIF.org, researchers can locate, download, and freely use a large volume of data showing where occurrences of vectors, hosts, and reservoir species have been recorded (through human observations, specimens, DNA sampling, and other sources of evidence) and shared by the GBIF network of data publishers.

Links between biodiversity and human health are complex and they occur from microorganism to ecosystem levels with the interplay of drivers at different scales, i.e. local, regional and global, and acting over different lengths of time [6]. Species occurrence data available through GBIF contribute to understanding these complex links and support the production of various models for risk prediction and assessment, vector and pathogen spread, among other uses.

The World Health Organization (WHO) emphasizes the links to wildlife data in its Road Map for neglected tropical diseases 2021–2030 [7], and calls for additional efforts to streamline the complementary biodiversity data on human diseases in the Global Vector Control Response 2017–2030 noting specifically that [8]:

> *'Entomological, epidemiological and intervention data are often managed separately without linkage, resulting in insufficient information on the impact of vector control interventions on entomological parameters and pathogen transmission... Database development and management experience is needed to ensure linkage of entomological, epidemiological and intervention data into a comprehensive monitoring and evaluation platform that ideally incorporates geo-referencing. Skills in information and communication technologies as well as behavioural change communication and community and local authority engagement are also required.'*

To help improve the availability, access, and use of biodiversity data linked to human diseases, in 2020 the Secretariat of GBIF established a task group [9, 10] composed of experts on vectors and data science. The primary Task Group mandate was to: (i) assess, with the objective of exploring the current and potential application of GBIF data and its existing infrastructure in addressing research questions and policy related to vectors, hosts/reservoirs and pathogens, (ii) identify, how and where improvements could be made in the current data model and standards, and (iii) actively work on vector data mobilization



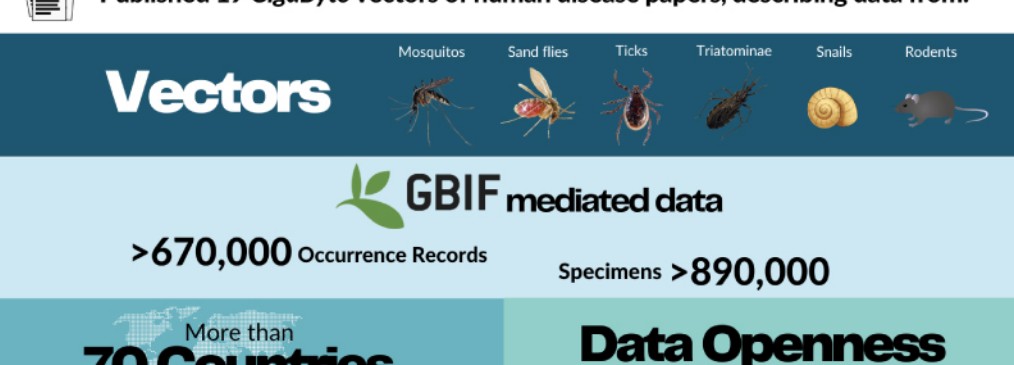

**Figure 1.** Summary of the data mobilized (>670,000 occurrence records) from the 19 papers published by *GigaByte* in the two calls for vectors of human disease papers.

efforts aiming to improve the coverage and representation of such data in an effort to support the variety of uses required and requested by this research community. The task group works not only to capture the best available experiences, information, and recommendations related to major data holders and users, but also to prioritize data targets by diseases, taxa, and regions. The task group is also considering the best means of organizing data necessary for addressing policy needs such as biodiversity indicators for IPBES (Intergovernmental Science-Policy Platform on Biodiversity and Ecosystem Services), the Kunming-Montreal Global Biodiversity Framework, the Essential Biodiversity Variables framework [11], and the Sustainable Development Goals (SDGs). Of which SDGs 14 and 15 being most relevant for life below water and on land, and also SDG3, which ensure healthy lives and promote well-being for all at all ages and its relevant targets, as well as the One Health and Planetary Health agendas.

Here, we aim to highlight the significance of data mobilization, the publication of data papers, data sharing, and open science in vector-borne disease research through a series of 19 papers published under a collaborative call by the TDR/WHO and GBIF. The first call resulted in the publication of 11 papers [12] and now we present the eight data papers that were published in the second call [13] (Figure 1).

In addition to improving coverage and use of GBIF-mediated data today, an important consideration of the task group is the requirements to identify data mobilization targets for species occurrences and other types of data on non-human species relating to human diseases, such as species interactions, traits, sequence- and multimedia, and links to external biomedical and human data, as long as they can be accessed respecting all ethical recommendations and processes. This does not presuppose that all data should be directly accessible through GBIF.org, but may suggest future functionalities clearly linking data shared through GBIF with other data sources [14].

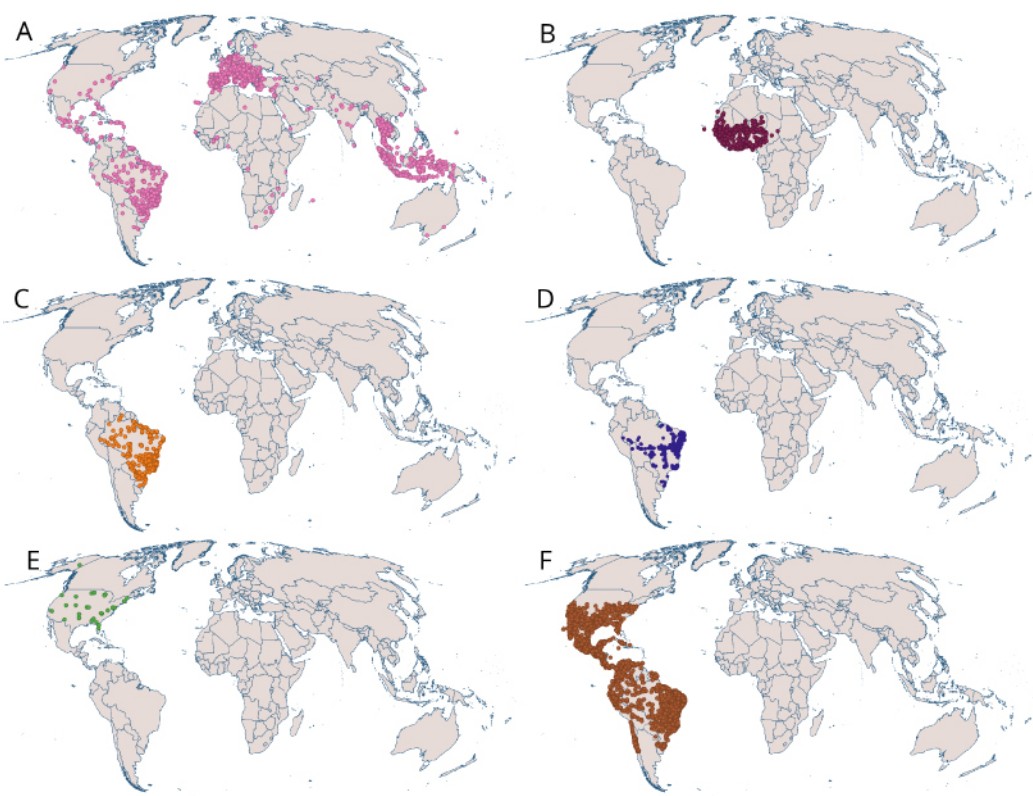

**Figure 2.** Map showing the distribution of georeferenced observations from both calls for data papers (2022–2023). The datasets included are all hosted via GBIF [22–43]. (A) Mosquitoes; (B) Rodents; (C) Sand flies; (D) Snail; (E) Ticks; (F) Kissing bugs.

As part of the efforts of the task group to mobilize data on vectors in 2023, TDR , GigaScience Press and GBIF renewed their partnership for a special issue focused on publishing new datasets that present biodiversity data for research on vectors of diseases. This call mobilized more than 500,000 occurrence records and 675,000 specimens from more than 50 countries [13], the distribution of georeferenced observations from both calls for data papers (2022–2023) can be visualized in Figure 2.

In recent years, the importance of data publication, data sharing, and open science has gained significant attention in the health sector [15, 16]. The TDR has been promoting open science and supporting the FAIR (Findable, Accessible, Interoperable, and Reusable) data principles, sharing access to scientific research and data, recognizing the potential for accelerated progress through collaboration, knowledge exchange, and most importantly, to support integration of research output into public health systems and policies at local, national, regional and global levels [17]. Furthermore, scientific journals such as *GigaByte* have actively supported this movement by emphasizing the publication of data papers with open data and encouraging transparency in methodology. Additionally, the GBIF has been facilitating biodiversity data sharing and providing the infrastructure for researchers to access and contribute to global datasets related to health [18].

The eight papers published in the second call for data papers (plus this one) build on the number of vector records available under FAIR principles on GBIF. These papers present

data on mosquitoes, snails and rodents, covering more than 171,622 records linked to 216,177 specimens across 63 countries and 5 continents (and particularly increase the data coverage in Asia and Africa). The data described in the contributed papers includes occurrence records and sampling events (systematic collections instances with documented sampling effort). Similar to the initial series, the papers are all connected to preprints, and many of them offer multilingual options, enhancing accessibility for French and Spanish speakers.

Open data in vector research holds advantages in better understanding and tackling the pathogens they transmit. Enhanced surveillance, modeling, and transboundary collaboration are among some of the advantages of FAIR open data on vectors, hosts and the pathogens. Papers in the first series made contributions to modeling vector/disease risk, to train machine-learning detection and classification of mosquitoes using routines on linked images collected from an app [19]; a dataset of tick abundance, diversity and pathogen infection was promoting One Health approaches to understanding transmission risk and improving surveillance [20]; and a comprehensive kissing bug (Triatominae) database with updates on the taxonomy, temporal and geographical species records. The dataset collates data from disparate sources and makes it available through GBIF, revealing data on the biodiversity and epidemiology of different vector-borne diseases, e.g., Chagas disease, to a wider audience [21].

## IMPORTANT STORIES THAT CAN BE TOLD THROUGH OPEN DATA

This second call mobilized a smaller number of records, but a wider geographic and taxonomic coverage of data on vectors was achieved. The papers contain interesting stories about mosquito, snail (Planorbidae), and rodent data that will be useful to the broader community.

Malaria continues to be a significant vector-borne disease in the tropical regions, affecting nearly 250 million individuals and resulting in over 600,000 deaths each year [44]. The importance of anopheline mosquitoes (Diptera: Culicidae) is reflected in the four papers presenting data on their occurrence from all continents except Antarctica [45–48].

The first paper presents data on mosquitoes collected for the project REACT ('Insecticide resistance management in Burkina Faso and Côte d'Ivoire: research on vector control strategies') under a randomized controlled trial. More than 60,000 *Anopheles* records are published as a Darwin Core archive in GBIF containing data for events, occurrences, identification methods, and environmental data [45].

The second paper describes the Bonne-Wepster subcollection which is held by the Naturalis Biodiversity Center in Leiden, it has 52,102 records of mosquitoes collected mainly in the former Dutch colonies of Indonesia and Suriname by medical entomologist Johanna Bonne-Wepster (Figure 2). Additional mosquito records from the former Rijksmuseum van Natuurlijk Historie collection and the former Zoölogisch Museum Amsterdam Nederland collection were also digitized and added to the dataset, totalling 55,706 records. The digitization of the Bonne-Webster subcollection was an effort to preserve information on the specimens, some over 100 years old, as associated data in field books were deteriorating due to old age. Also, the field data of the collection originally did not contain any coordinates, and efforts to georeference the 'old' data albeit very challenging now provide baseline data on mosquitoes collected in the early 20th century, thus turning these 100 year

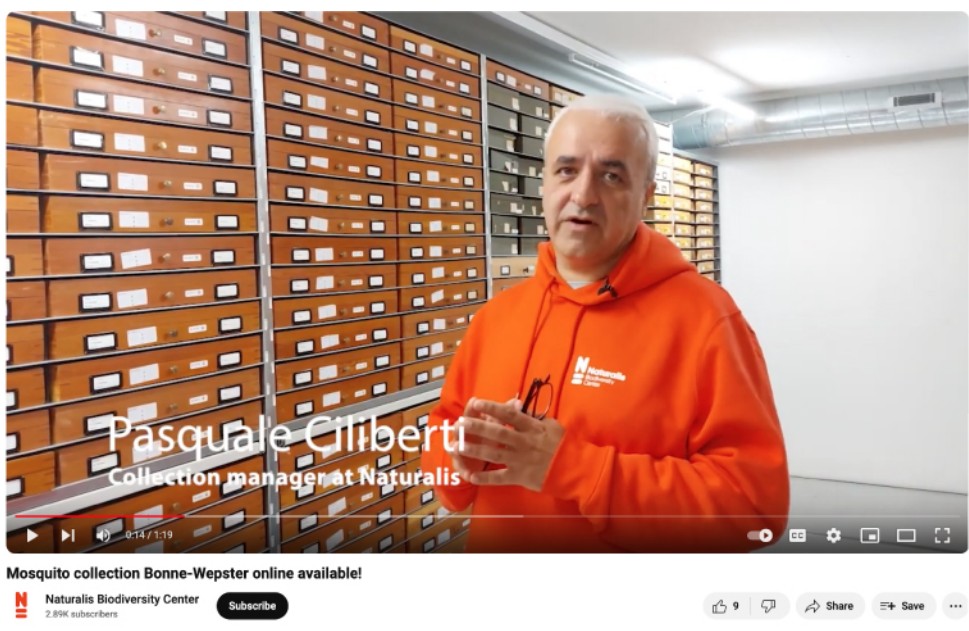

**Figure 3.** Video overview of the Bonne-Wepster Mosquito collection from some of the curators and authors involved in digitisation [47]. https://www.youtube.com/watch?v=6m4pXCScPDw

old and nearly lost field records into 21st century digital records giving a georeferenced baseline of species occurrences in the early 20th century (Figure 3) [46].

Another paper describing mosquito fauna comes from Thailand with a curated dataset collated from work on routine surveillance carried out by the Thailand Malaria Elimination Program database [48] maintained by the Division of Vector Borne Diseases, Department of Disease Control, Ministry of Public Health, as well as research projects from various Thai universities, including Department of Entomology, Faculty of Agriculture, Kasetsart University. Mosquitoes were identified by morphology and molecular techniques and the dataset comprises 12,278 records of at least 117 mosquito species and it covers 1,725 locations across 66 (out of 77) Thai provinces [49].

The fourth paper describes the anopheline fauna in Kinshasa, including molecular typing to identify components of the *Anopheles gambiae* complex which was widespread throughout Kinshasa. Two species of the complex are present in Kinshasa: *Anopheles gambiae* and *Anopheles coluzzii*, with *An. gambiae* being the most prevalent [50].

In many countries, governmental institutes perform entomological surveillance by carrying out collections of arthropod vectors. In tropical areas of the world, *Aedes aegypti* mosquitoes are responsible for the transmission of many arboviral diseases. This mosquito species is highly adapted to human environments breeding in a variety of containers that sustain larval development.

In a paper from Colombia, larvae of *Aedes aegypti* were sampled in endemic areas for dengue. A dataset with records of 3,806 immature pupae of *Ae. aegypti* was published in GBIF, and the paper suggests that it is possible to mitigate vector-borne diseases using vector spatial patterns [51]. This paper is a follow-up on a data paper published in the first series on the distribution of Culicidae from Cauca Department, Colombia [52].

Another paper presents data from 6,943 observations of adult *Aedes aegypti* and *Aedes albopictus* from the Democratic Republic of Congo. Samples were collected in a chikungunya post-epidemic zone in the city of Kinshasa, between 2020 and 2022. This study provides information on the ecology and phenology of *Aedes* mosquitoes, showing a predominance of *Ae. albopictus* over *Ae. aegypti* and increased density during the rainy season, pointing to a higher dengue fever transmission rate during this period [53].

A dataset was published containing data from 127 trapping studies conducted in 14 West African countries between 1964 and 2022 on rodents, an important group of reservoirs/hosts of diverse microorganisms that can be pathogenic to humans and non-human animals. The dataset includes data for 65,628 individual small mammals identified to the species level from over 1,600 trapping sites and it also includes 32 microorganisms, identified to the species or genus levels, that are known or potential pathogens [54].

And one paper was published with data from Collection of Mollusks of the Oswaldo Cruz Institute (FIOCRUZ/CMIOC) on the family Planorbidae, a group of freshwater gastropods that includes the medically important genus *Biomphalaria* that acts as intermediate host of parasitic flatworms called schistosomes. The dataset has 7,267 lots originating from 55 countries, distributed in 20 genera and 75 species collected from 1948–2023. *Biomphalaria* data covers 42 countries, and these records can contribute to a better understanding of the distribution of schistosomiasis and its transmission [55].

Unlike the first call, in which a thematic help desk service was available, this second call had help desk services provided either by the National Nodes [SiB Colombia, SiBBr/Brazil, GBIF France, Netherlands Biodiversity Information Facility (NLBIF), National Biodiversity Network (NBN/UK)], by GBIF's regional support teams in Africa and Asia, task group members (DS, KI, PS), and/or by GBIF's help desk. The GigaScience Press curation team assisted with the curation of other supplemental materials (including protocols) and carried out data audit and review on all of the submissions, they also acted as provisional data publishers in GBIF while the original institutions were awaiting endorsement for two submissions, and they are currently still the publishing institution for one dataset hosted by The National Biodiversity Network/UK.

Challenges and issues encountered varied from issues in registering institutions and finding publishers to technical challenges such as biotic interaction data mapping to Darwin Core Standard from the rodent dataset [54]. These technical issues were creatively solved by using the terms dwc:basisOfRecord to distinguish records of host and pathogen, described as HumanObservation and MaterialSample, respectively; dwc:eventID and dwc:parentEventID to connect records of hosts and pathogens; dwc:occurrenceRemarks has been used to describe the number of individual hosts that were assayed for pathogen detection; and dwc:occurrenceStatus for presence/absence of the pathogen.

It is of special interest to mention that apart from the use of extensions, such as the ResourceRelationship extension, the current data model in GBIF does not adequately support the publication of interaction data, i.e. herbivory, pollination, parasitism, and some degree of data loss does occur. But in order to support publication of more complex data, such as interaction data and other types of data, GBIF along with the Darwin Core Maintenance Group, is working towards extending Darwin Core Archives capabilities to allow the publication of more complex types of biodiversity data without data loss and an improved data model is expected in the near future [56].



## CONCLUSION

The two sponsored calls for papers are now over, but the series will remain open for submissions and future papers that meet the scope will be added to the series page. *GigaByte* will continue to provide a one-stop home for discovery and discussions around this critically important data type.

This publication was supported by the GBIF health task group, which consists of experts from various fields including epidemiology, public health, ecology, medical entomology, biodiversity informatics, and data science, and its members, acknowledging that human and animal health, climate, biodiversity and environment are interlinked. In order to tackle disease control and possibly elimination, stakeholders must have access to a wide range of data openly shared under the FAIR principles, to support early detection, analyze and evaluate, and to be able to inform policy improvements and/or development.

The task group will continue to engage in communication and capacity building efforts to raise awareness and will continue to propose training activities and collaborations to strengthen the use of GBIF platform among the health community. The task group will continue to work to improve representation of health-related data in GBIF all along its current mandate.

## EDITORS' NOTE

This Guest Editorial is part of a series of Data Release articles working with GBIF and supported by TDR, the Special Programme for Research and Training in Tropical Diseases hosted at the World Health Organization, in order to publish datasets on vectors of human diseases [13]. Whilst the sponsorship of the WHO is now over, the series remains open for submissions. The *GigaByte* Editors would like to thank the GBIF expert task group for their support and oversight, the crucial assistance of the GBIF helpdesk, and the financial support of TDR and the WHO who all helped make this series happen.

## DATA AVAILABILITY

All the observation data in described in the series is available via GBIF [22–43].

## ABBREVIATIONS

FAIR: Findable, Accessible, Interoperable, and Reusable; GBIF: Global Biodiversity Information Facility; SDG: Sustainable Development Goals; TDR: the Special Programme on Research and Training in Diseases of Poverty; VBD: vector-borne disease; WHO: World Health Organization.

## DECLARATIONS

### Ethics approval and consent to participate

The authors declare that ethical approval was not required for this type of research.

### Competing interests

The authors declare that they have no competing interests.

### Funding

This series was supported by sponsorship from the TDR/WHO. TDR, the Special Programme for Research and Training in Tropical Diseases co-sponsored by UNICEF, UNDP, the World

Bank and WHO, is able to conduct its work thanks to the commitment and support from a variety of funders. These include our long-term core contributors from national governments and international institutions, as well as designated funding for specific projects within our current priorities. For the full list of TDR donors, please visit our website at: https://www.who.int/tdr/about/funding/en/. QG acknowledges funding of the European Union's Horizon Europe Research and Innovation Programme (ID No 101059592) through the B3 (Biodiversity Building Blocks for policy) project.

## Acknowledgements

We would like to thank all the authors of papers from the second series on disease vectors in *GigaByte.* We also would like to express our gratitude to the support provided by SiB Colombia, SiBBr/Brazil, GBIF France, Netherlands Biodiversity Information Facility (NLBIF), National Biodiversity Network (NBN/UK), GBIF's regional support teams in Africa and Asia, GBIF's help desk and the GigaScience Press curation team.

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
