## [Editor Report]

Editor’s AssessmentClosing the two sponsored calls for papers in the GigaByte Vectors of Human Disease series (see https://doi.org/10.46471/GIGABYTE_SERIES_0002) this Editorial provides an overview and lessons learned of how the targeted data publication scheme went. Written by the GBIF/TDR health taskforce members who helped oversee the series call and publication. The series to this point has led to the sharing of over 670,000 occurrence records linked to over 890,000 specimens. The two sponsored calls for papers are now over, but the series will remain open for submissions, and the task group will continue to work to improve representation of health-related data in GBIF.

---

## [Reviewer Report]

Reviewer name and names of any other individual's who aided in reviewer Scott C EdmundsDo you understand and agree to our policy of having open and named reviews, and having your review included with the published papers. (If no, please inform the editor that you cannot review this manuscript.)YesIs the language of sufficient quality?YesPlease add additional comments on language quality to clarify if needed
Are all data available and do they match the descriptions in the paper? YesAdditional CommentsAre the data and metadata consistent with relevant minimum information or reporting standards? See GigaDB checklists for examples <a href="http://gigadb.org/site/guide" target="_blank">http://gigadb.org/site/guide</a>YesAdditional CommentsIs the data acquisition clear, complete and methodologically sound?YesAdditional CommentsIs there sufficient detail in the methods and data-processing steps to allow reproduction?YesAdditional CommentsIs there sufficient data validation and statistical analyses of data quality? YesAdditional CommentsIs the validation suitable for this type of data?YesAdditional CommentsIs there sufficient information for others to reuse this dataset or integrate it with other data?YesAdditional CommentsAny Additional Overall Comments to the AuthorThis is a commissioned Guest Editorial by the GBIF Task Group members overseeing the vectors of human disease series recently published in GigaByte. As such it does not require peer-review. This has been copy-edited by the Editorial team, and as well as some suggested edits to the text and references an Editors Note has been added to better explain the history of this commissioned paper. Additional suggestions have been made about the figures and they are now being revised in this revision process.RecommendationAccept